# Quantifying Heart Rate Changes After Delta-9-Tetrahydrocannabinol Administration Using a PBPK-PD Model in Healthy Adults

**DOI:** 10.3390/pharmaceutics17020237

**Published:** 2025-02-12

**Authors:** Lixuan Qian, Zhu Zhou

**Affiliations:** Department of Chemistry, York College, City University of New York, Jamaica, NY 11451, USA

**Keywords:** delta-9-tetrahydrocannabinol, heart rate, pharmacodynamics

## Abstract

**Background**: As cannabis becomes legal in several U.S. states, the risk of THC-induced tachycardia increases. This study aimed to develop and verify a physiologically based pharmacokinetic–pharmacodynamic (PBPK-PD) model to assess the impact of THC and its active metabolite, 11-hydroxy-THC (11-OH-THC), on the heart rate of healthy adults. **Methods**: A PBPK-PD model for intravenous (IV) 11-OH-THC administration was first developed. Secondly, a PBPK-PD model for IV THC, combined with the metabolized 11-OH-THC, was established, verified, and validated. Direct PD models driven by the plasma, brain, and heart concentrations of THC and 11-OH-THC predicted using our previously verified PBPK model were tested for model development. Finally, the risks of tachycardia at a rest condition from various doses of oral and inhaled THC were simulated for 500 individuals aged 18–65 years, with a sex ratio of 1:1 and a baseline heart rate of 70 beats per minute. **Results**: The PD model was best described by a direct nonlinear E_max_ model driven by the sum of the total THC and 11-OH-THC concentrations in their effect compartments linked to their heart compartments. In 42 simulated dosing regimens with THC doses ranging from 2 to 69.4 mg, 97% of the observed heart rates or heart rate changes following THC administration fell within the 5th to 95th percentiles of the model-predicted values. Similarly, for two simulated 11-OH-THC IV doses, 93% of the observations fell within this range. Simulations indicated that half of the simulated population would experience tachycardia at doses of 60 mg and 15 mg of THC for oral and inhaled administration, respectively. The simulated risks of tachycardia based on specific conditions should be interpreted with caution. **Conclusions**: Our verified PBPK-PD model successfully describes the heart rate changes in healthy adults after IV, oral, and inhaled THC administration. This model provides a tool to predict the effects of THC and its primary metabolite on heart rates, offering valuable insights for assessing the risk of tachycardia in both clinical and recreational cannabis use.

## 1. Introduction

Delta-9-tetrahydrocannabinol (THC) is a prominent cannabinoid isolated from the plant *Cannabis sativa* (cannabis) [1]. The synthetic form of THC has been approved for treating nausea and vomiting associated with cancer treatments and HIV-induced anorexia [2]. Recently, the recreational use of cannabis has been legalized in some states of the United States (U.S.) [3]. In 2018, U.S. federal guidelines established that plant material containing 0.3% or less THC is classified as a legal agricultural commodity [4]. However, the average THC concentration in illegal cannabis samples increased from approximately 4% in 1995 to approximately 13% in 2022, thereby increasing the chance of ingesting substantial amounts of THC [5,6]. Its increasing potency and widespread use raise concerns about cardiovascular effects, including tachycardia, an adverse effect that is considered a predictor of cardiovascular morbidity and mortality [7,8,9].

THC exerts its physiological effects primarily by binding to cannabinoid receptors type 1 (CB1) and CB2 [10]. The primary metabolite of THC is 11-hydroxy-THC (11-OH-THC), which shared a similar binding affinity for CB1 and CB2 receptors in an in vitro study [11], and exhibited comparable potency in activating these receptors in an animal study [12]. Both THC and 11-OH-THC have been reported to induce tachycardia [7,9,13,14,15]. While early reports suggested that cardiac death after THC use was associated with pre-existing cardiac conditions, and several cases of sudden cardiac death in healthy young adults have highlighted the need for further understanding of THC’s effect on heart rate [16].

The acute heart rate increases following THC administration are considered to be a main factor in THC-induced sudden cardiac death, and the effect is primarily linked to CB1 receptor activation [9]. The exact mechanism responsible for tachycardia remains unclear due to the extensive distribution of CB1 receptors across the heart, the sympathetic and parasympathetic nervous systems, and the central nervous system (CNS) [17,18]. Some studies have suggested that THC-induced CB1 receptor activation in the CNS is the primary cause of heart rate changes [19]. However, in rat studies, cannabinoids injected into the rostral ventrolateral medulla, a region of the brain involved in cardiovascular regulation, produced different effects, including increases, decreases, or no change in heart rate [9]. Additionally, the peak heart rate change occurred earlier than the peak of other CNS responses in the brain, suggesting that the CB1 receptor activation in the CNS may not be directly responsible for the acute increase in heart rate after THC administration [20]. Several clinical studies have suggested that THC’s effect on autonomic nerves may induce changes in the heart rate of humans [21,22]. In contrast, the findings of some animal studies provide evidence that THC does not affect heart rate via the autonomic ganglia [23,24]. Experiments conducted on isolated rat hearts have shown that THC can moderately increase heart rate, while THC administration in rats generally leads to a reduction in heart rates [25].

Given the physiological effects of THC on heart rate, particularly its potential to induce tachycardia, it is crucial to understand how THC changes the heart rate. Many clinical trials have provided valuable insights, and quantitative modeling techniques, such as population pharmacokinetics (popPK), have been used to examine the relationship between THC dose and heart rate changes [20,26]. However, these models have notable limitations. They fail to account for the synergistic effects of 11-OH-THC, and only the heart rate changes driven by THC plasma concentration were tested. Additionally, extrapolation for the popPK model is limited by the modeling population used in model development, making it challenging to predict heart rate changes across various dosages, routes of administration, and population demographics. In contrast, by incorporating physiological and biological knowledge at the organism level, along with the physicochemical properties of the drug, physiologically based pharmacokinetic (PBPK) modeling allows for a more accurate description of drug exposure in various organs, including the heart [27]. Through simulating local concentrations at the possible site of action, PBPK modeling is better suited to extrapolate pharmacokinetics (PKs) to various doses and populations. Additionally, unlike previous models that focus solely on THC, PBPK models can incorporate the effects of 11-OH-THC, enabling the prediction of 11-OH-THC concentration in the related organs and establishing the exposure–response relationship between 11-OH-THC and heart rate changes.

This research aims to address existing gaps by developing and verifying a PBPK-pharmacodynamic (PD) model to predict heart rate changes caused by THC and 11-OH-THC. Additionally, the model will assess the risk of tachycardia at rest condition after different THC doses administered via oral and inhaled administration routes in healthy young adults.

## 2. Materials and Methods

### 2.1. PBPK-PD Software, Data Acquisition, and Parameter Assessment

PBPK-PD models were developed using the Simcyp™ PBPK Simulator (version 22, Certara, Sheffield, UK). A virtual healthy adult population was used during model development. Ten trials were simulated using study designs that closely matched the corresponding clinical trials and ensured consistency in the heart rate baseline, the dosing regimen, the number of subjects, the age range, and the proportion of females. All accessible clinical data, including THC concentration–time profiles and heart rate–time profiles or heart rate change percentile–time profiles, were sourced from the published literature or digitized using WebPlotDigitizer version 4.

### 2.2. Datasets for Model Development

The inclusion criteria for THC and 11-OH-THC clinical studies used for the PBPK-PD model development, verification, and validation were (1) the studies involved healthy adult participants; (2) heart rate–time profiles or heart rate change percentile–time profiles were provided; (3) dosing and/or concentration–time information of THC were provided; and (4) THC was administered via the intravenous (IV), oral, or inhalation routes. Exclusion criteria were (1) an unspecified THC dose; (2) a non-zero baseline THC concentration at the start of the clinical trial; (3) a lack of unit for heart rate changes; (4) the absence of a baseline heart rate when changes were reported in beats per minute (bpm); and (5) the reporting of heart rate changes based on the control group. In total, the observed data used to develop and verify the THC PBPK-PD model for heart rate changes were derived from 30 published clinical trials comprising 42 dosing regimens [14,15,28,29,30,31,32,33,34,35,36,37,38,39,40,41,42,43,44,45,46,47,48,49,50,51,52,53,54,55]. An IV clinical trial administering both THC and 11-OH-THC was used to develop the PBPK-PD model. The parameter optimizations and the model assumptions were verified using data from one inhalation THC study and one oral THC study. Studies used for model verification and validation are presented in Appendix A. Heart rate time points after THC doses that were lower than the baseline heart rate were excluded from model development, verification, and validation.

### 2.3. PBPK Model Optimization

A PBPK-PD model of THC was constructed using the Simcyp™ Simulator (version 22), as illustrated in Figure 1. Details of the PBPK model inputs for THC have been previously described [56]. The model included IV, oral, and inhaled THC administrations. The disposition of both THC and 11-OH-THC was described using a whole-body PBPK model with predicted tissue partition coefficients. THC is not reported to be a substrate of transporters [57]. The distribution for tissues, including the brain and heart, was assumed to be perfusion-limited.

The IV and oral clinical trials used in this study did not provide concentration–time profiles for THC or 11-OH-THC. Therefore, the previously developed PBPK models for IV and oral THC were used without modification [56]. For inhaled THC trials, three types of clinical trials were included: clinical trials with THC concentration–time profiles included in our previous THC PBPK study (type 1) [56]; clinical trials with THC concentration–time profiles not included in our previous study (type 2); and clinical trials that provided only dosing information without THC concentration–time profiles (type 3). The absorption parameters for first-order lung absorption were modified based on the aforementioned data type, while other components of the PBPK model remained consistent with our previously published model [56].

For type 1 inhalation studies, the lung absorption parameters were the same as those reported in our published PBPK model [56], and THC plasma concentration profiles were well captured. For each type 2 inhalation study, the fraction of the drug absorbed from the lung (lung f_a_) and the first-order absorption rate constant for the lung (lung k_a_) were optimized with the same method as in our published THC PBPK study to accurately predict the observed THC concentration–time profiles [56]. For type 3 studies, lung f_a_ and lung k_a_ values from our published THC PBPK studies or those optimized from type 2 studies, which predicted the peak heart rate most closely matching the observed peak heart rate, were selected [56]. Identical THC inhaled-absorption parameters were used for all dosing regimens within the same type 3 clinical study. Specifically, one dosing regimen was used for the model optimization of lung f_a_ and lung k_a_, while the other dosing regimens used the same absorption parameters during validation. All the absorption parameters used are summarized in Table 1.

### 2.4. PBPK-PD Model Development

The workflow for PD model development is summarized in Figure 1. The PD model for THC and 11-OH-THC was described using a direct effect model, coded using the custom Lua Models within the Simcyp™ simulator. The structural model was modified based on Strougo et al. [20]. We tested the increase in the heart rate driven by the total THC and 11-OH-THC concentration in the effect compartment associated with different THC compartments (plasma, brain, or heart compartments).

The heart rate model driven by 11-OH-THC was first established and verified using a nonlinear maximum effect (E_max_) model with an effect compartment (Equations (1) and (2)):(1)$$\frac{dC_{11-OH-THC,effect}}{dt}=k_{1e,11-OH-THC}\times C_{11-OH-THC}-{k_{e0,11-OH-THC}\times C}_{11-OH-THC,effect}$$(2)$$∆\;Heart\;rate\;\left(\%\right)=E_{max}\times \frac{C_{11-OH-THC,effect}}{EC_{50}+C_{11-OH-THC,effect}}\times 100\%$$
where $C_{11-OH-THC,effect}$ is the 11-OH-THC concentration in the effect compartment associated with the compartment driving the heart rate changes; $k_{1e,11-OH-THC}$ is the rate constant for 11-OH-THC transport from the compartment driving heart rate changes to its effect compartment; $C_{11-OH-THC}$ is the total 11-OH-THC concentration in the tested THC compartment; $k_{e0,11-OH-THC}$ is the elimination rate constant of the 11-OH-THC effect compartment; ∆ Heart rate (%) is the percentage of heart rate changes from the baseline; $E_{max}$ is the maximum fractional increase from the baseline heart rate induced by 11-OH-THC; and $EC_{50}$ is the 11-OH-THC concentration required to obtain 50% of the maximum change. k_1e_, k_e0_, E_max_, and EC_50_ were estimated using the Simcyp™ Parameter Estimation module and clinical data from 11-OH-THC IV studies to best capture heart rate changes induced by 11-OH-THC. k_1e_ and k_e0_ were assumed to be the same unless the optimized values could not capture the observations [58]. The interindividual variabilities of E_max_ and EC_50_ used in our model were derived from a published THC heart rate popPK/PD model [20].

Next, the heart rate model driven by THC and 11-OH-THC was established. The 11-OH-THC effect compartment model was the same as above. We assumed that the EC_50_ and E_max_ of THC and 11-OH-THC were the same according to an animal study [12]. Because of the same binding affinity of THC and 11-OH-THC to CB1, the heart rate models after THC administration were assumed to be driven by the sum of total THC and 11-OH-THC concentrations in their effect compartment (Equations (1) and (3)–(5)):(3)$$\frac{dC_{THC,effect}}{dt}=k_{1e,THC}\times C_{THC}-{k_{e0,THC}\times C}_{THC,effect}$$(4)$$Heart\;rate\;(bpm)=Baseline\times \left(1+E_{max}\times \frac{C_{THC,effect}+C_{11-OH-THC,effect}}{EC_{50}+\left(C_{THC,effect}+C_{11-OH-THC,effect}\right)}\right)$$(5)$$∆\;Heart\;rate\;\left(\%\right)=E_{max}\times \frac{C_{THC,effect}+C_{11-OH-THC,effect}}{EC_{50}+\left(C_{THC,effect}+C_{11-OH-THC,effect}\right)}\times 100\%$$
where $C_{THC,effect}$ is the total THC concentration in the effect compartment linked to the compartment driving the heart rate changes; $k_{1e,THC}$ is the rate constant for THC transport from the compartment that drives heart rate changes to the THC effect compartment; $C_{THC}$ is the total THC concentration in the tested THC compartment; $k_{e0,THC}$ is the elimination rate constant of the THC effect compartment; Baseline is the baseline of the heart rate, which is the observed heart rate from the studies measured before THC dosing; $E_{max}$ is the maximum fractional increase from the baseline heart rate induced by THC and 11-OH-THC; and $EC_{50}$ is the total THC and 11-OH-THC concentration in the effect compartment required to obtain 50% of the maximum change. If the heart rate (bpm) was reported, then Equation (4) would be used, with the closest observed mean heart rate before THC administration serving as the baseline. If the percentiles of heart rate changes from the baseline were reported, Equation (5) would be used.

### 2.5. Model Verification and Validation

Model verification of the parameter estimations and the assumption that the THC compartment drives the effect was performed using two clinical trials with intensive heart rate sampling points: one study involving oral administration, and the other involving inhaled administration [30,50]. Model validation to assess the model’s performance was also conducted, with the validation dataset including all other studies listed in Appendix A. The workflow is shown in Figure 1. For all studies, the prediction performance was assessed by determining whether the observed heart rates fell within the 5th to 95th percentiles of the predicted values. For the dosing regiments whose absorption parameters were not optimized with peak heart rate change, prediction performance was also evaluated by checking whether the predicted peak heart rate or peak heart rate change percentile fell within a 1.3-fold range of the observed values [59]. Typically, prediction performance for PBPK models is assessed based on whether predictions fall within a two-fold range of the observations. However, in our case, the baselines for most studies were within two-fold of the maximum observed heart rate. Therefore, we used a more stringent criterion of 1.3-fold instead of two-fold.

### 2.6. Simulation of Risk of Tachycardia at Rest Condition

The potential risks of tachycardia at rest condition caused by THC were simulated using the verified THC-heart rate PBPK-PD model, with THC doses ranging from 1 mg to 200 mg, administered orally or by inhalation. The therapeutic doses of oral THC (5 mg and 15 mg), and inhaled THC corresponding to three doses of legal cannabis cigarettes (1, 5, and 10 cigarettes, assuming a THC concentration of 0.3% and a cigarette weight of 850 mg [60], equating to 2.55 mg of THC per cigarette), were also simulated. In each simulation, the heart rates of 25 healthy adults between the ages of 18 and 65 were simulated across 20 trials following a single THC dose. The baseline heart rate was set at 70 bpm, and the sex ratio was assumed to be 1:1. We chose 70 bpm as the baseline for the simulation based on a recent guideline suggesting that the resting heart rate in adults ranges from 60 to 80 bpm [61]. A tachycardia event is defined as a simulated peak heart rate exceeding 100 bpm in an individual [62]. The risk of tachycardia after THC administration in each trial was described by the following equation (Equation (6)):(6)$$\mathrm{Risk}\left(\%\right)=\frac{\mathrm{Number}\;\mathrm{of}\;\mathrm{individuals}\;\mathrm{with}\;\mathrm{peak}\;\mathrm{heart}\;\mathrm{rate}\;>\;100\;\mathrm{bpm}}{\mathrm{Number}\;\mathrm{of}\;\mathrm{individuals}\;\mathrm{simulated}}\times 100\%$$

A sensitivity analysis of the THC heart rate PD model was conducted to assess the impact of PD parameters on the risk of tachycardia after 60 mg of THC oral administration.

## 3. Results

The plasma concentration–time profiles for IV and orally administered THC were not reported in the studies. The PBPK model of THC, with optimized inhalation absorption parameters (Table 1), captured the plasma concentration–time profile of THC well. The predicted versus observed concentration–time profiles for inhaled THC are shown in Appendix A. Approximately 99% of the observed THC concentrations fell within the 5th to 95th percentiles of the prediction. The mean THC concentrations in the plasma, brain, and heart compartments after IV, oral, and inhaled THC administration for the modeling dataset and the verification dataset are presented in Appendix A. The predicted brain C_max_ was approximately 3–4 times the predicted plasma C_max_, which is consistent with the observed range of 2–5 times in post-mortem humans [63].

The THC-induced increase in heart rate was predicted with the greatest accuracy when driven by concentrations in the heart compartment. When driven by the THC plasma concentrations, verification showed that the model can capture the heart rate changes after IV and oral THC administration. However, after inhaled THC administration, the peak heart rate changes were under-predicted. When driven by THC brain compartment concentrations, the model can capture the heart rate changes following IV and inhaled THC administration but under-predicts the heart rate changes after oral administration. The Lua codes used for developing the PD model are presented in the Appendix A.

The predicted pharmacological effect of the developed PBPK-PD model was found to be in good agreement with the clinical observations. Figure 2 compares the model-predicted peak heart rates or heart rate changes for THC and 11-OH-THC against the observed values. The E_max_, EC_50_, k_1e,11-OH-THC_, and k_e0,11-OH-THC_ values optimized for the 11-OH-THC and the optimized K_1e,THC_ and K_e0,THC_ values for the THC effect compartment are listed in Table 1. A detailed comparison between the predicted and the observed heart rate changes for 11-OH-THC IV studies is shown in Appendix A and Figure 3. The PBPK-PD model successfully captured the heart rate changes after IV, oral, and inhaled THC administration, and the heart rate changes after IV 11-OH-THC administration. In all trials included in the verification datasets, the predicted peak heart rate was within 1.3-fold of the observed peak (Figure 2). The ratio of the predicted to observed mean peak heart rate (R_max_) ranged from 0.97 to 1.05 for IV, 0.89 to 1.09 for oral, and 0.77 to 1.16 for inhaled administration. Furthermore, 97% and 93% of the observed heart rates fell within the 5th to 95th percentile of the predictions after THC and 11-OH-THC administration, respectively (examples in Figure 3).

The risks of tachycardia following oral and inhaled THC administration at a rest condition, ranging from 1 mg to 200 mg, were simulated. The dose–response relationship between THC dose and the risk of tachycardia for both oral and inhaled administration is summarized in Figure 4. For healthy young adults with a baseline heart rate of 70 bpm, 50% of the population was predicted to experience tachycardia at a dose of 60 mg for oral THC and 15 mg for inhaled THC at a rest condition. The risk of tachycardia for therapeutic doses of oral THC and for three doses of cannabis cigarettes is presented in Figure 4. The results of the sensitivity analysis are presented in Appendix A. E_max_ has the strongest impact on the risk of tachycardia risk, followed by EC_50_.

## 4. Discussion

Tachycardia has been well documented as a life-threatening adverse effect of THC, with fatalities recorded in patients with pre-existing cardiac disease and healthy young adults [9]. Despite these studies, significant gaps remain in understanding the mechanisms driving heart rate changes following THC administration. In this study, we address these gaps by developing and verifying PBPK-PD models for THC and its key metabolite, 11-OH-THC, which induced heart rate changes after administration via different routes. To the best of our knowledge, this is the first study to investigate the effects of THC on heart rate using THC concentrations at the site of action, the first to investigate the combined effect of THC and 11-OH-THC on heart rate, and the first study to assess the risk of tachycardia across various THC doses in oral and inhaled administration.

As of now, no previous PBPK-PD models have been developed for THC and heart rate. Two popPK/PD models were developed for inhaled THC and heart rate in separate studies. Each of these models was able to capture the heart rate changes observed in their respective clinical studies, but they were limited to a single study. One study reported an E_max_ value 1.4-fold higher than the other (93 bpm vs. 65.3 bpm) [20,26]. The discrepancy in key parameters between these two models presents challenges for extrapolation. Additionally, these popPK/PD models driven by plasma THC concentrations only included inhaled THC administration and have not been verified in other formulations. In contrast, our published PBPK model was developed for a wide range of dosing regimens and various routes of administration. Based on our published THC PBPK model, the THC PBPK-PD model for heart rate changes we developed in this study successfully predicted heart rate changes across 42 THC dosing regimens with IV, inhaled, and oral THC administration and a dose range from 2 to 69.4 mg with the same PD parameters, demonstrating strong potential for extrapolation.

The relationship between THC and heart rate remains poorly understood, with the precise mechanisms underlying its effects still to be fully elucidated. As aforementioned, while it is known that the CB1 receptor triggers heart rate changes following THC administration, the exact mechanism has not yet been fully understood [17]. Our model suggests that heart THC and 11-OH-THC concentrations induce the PD effect on heart rate. The effect compartment rate constants for THC and 11-OH-THC were optimized, with different values being obtained. The reason for the difference in effect compartment rate constants remains unclear; the differences in their physical and chemical properties may be a contributing factor.

There are limitations to our study. First, a first-order absorption model was applied to oral and inhaled THC because of the limited understanding of the absorption mechanisms for THC. Optimizations were required for inhaled THC administration to capture the PK profiles for different inhalation processes and methods. When the THC concentration–time profiles are absent in inhalation studies, optimizing inhalation absorption parameters based on peak heart rate changes may reduce the reliability of the validation. Second, the mean heart rate before THC administration was used as the baseline, but its inter-individual variability was not included in our model. In most studies, the variability of the baseline heart rate was not reported, therefore only the mean heart rate before THC administration in each dose regimen can be used as the baseline. Third, all participants in the clinical trials used in this study were in the rest state; the heart rate in the control group with no THC administration (placebo effect) changed only slightly. Some studies did not provide the heart rate changes for the placebo group [14,15,47]. The placebo group exhibited a variety of changes in heart rate, including increases and decreases, in different studies [30,32]. The influence of the placebo effect on the THC treatment group is unclear. Therefore, our model did not adjust the observed heart rate changes via placebo effects. The variability of the placebo effect across studies may be due to the different study designs (e.g., diet, additional behavior measures, and permitted activities during the study period), and thus, it is difficult to differentiate the placebo effect from the THC effect on heart rate. Many factors can impact heart rate in real-world scenarios, including changing body position, eating a meal, or watching a movie. The placebo effect may not be negligible for real-world scenarios and should be considered in future models. Fourth, the simulated risk of tachycardia following THC administration was evaluated under the specific scenario of a healthy population aged from 18 to 65, a sex ratio of 1:1, and a baseline heart rate of 70 bpm at a rest condition. This scenario does not reflect real-world variability, and the results from these simulations should be interpreted with caution. Further data on diverse patient populations and different THC routes of administration would enhance the model’s applicability and allow for more accurate predictions in real-world settings. Finally, our model does not include the mechanism of the THC effect on heart rate. Although the final models suggested using the THC heart compartment as the compartment driving the effect, our model is an empirical model. The increased tolerance to THC observed in chronic users was not considered [54]. Some chronic cannabis users had a lower baseline heart rate and a milder response to THC [54]. Due to the non-zero baseline THC concentration in chronic users, our study excluded this group and included only occasional users [54]. Currently available in vivo and in vitro data do not support modeling either the mechanism underlying THC-induced heart rate changes or the development of tolerance to THC’s pharmacodynamic effects. Incorporating data from a broader range of populations, including chronic cannabis users and individuals with varying baseline cardiovascular conditions, would help refine the model’s predictions and increase its clinical relevance.

Future studies could explore incorporating a CB1 receptor occupancy model, a tolerance model for the THC effect, and a more comprehensive investigation of THC’s mechanisms of action through the CNS. The placebo effect in real-world scenarios should be considered. Although acute heart rate changes are well documented, long-term use of THC may be associated with chronic cardiovascular issues, thus further research is needed to fully understand the long-term impact of THC on heart health.

## 5. Conclusions

In conclusion, we developed and verified heart rate PBPK-PD models for THC and 11-OH-THC based on our previous THC PBPK model. Our models accurately capture heart rate changes following IV, oral, and inhaled routes of THC administration. The robustness of our model in predicting heart rate changes across multiple THC dosing regimens strengthens its potential as a tool for predicting heart rate changes after THC administration and could guide future studies or clinical decision-making. Our models offer a prediction of THC’s impact on heart rate and help assess tachycardia risk in THC users, ultimately improving the safety of medical and recreational cannabis and THC use.

## Figures and Tables

**Figure 1 pharmaceutics-17-00237-f001:**
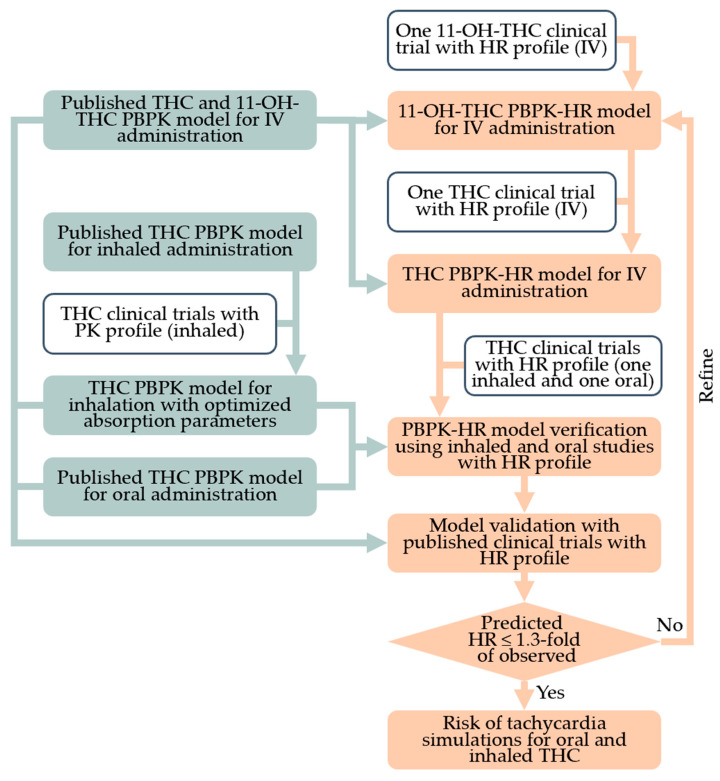
General workflow for delta-9-tetrahydrocannabinol (THC) physiologically based pharmacokinetic (PBPK)-pharmacodynamic (PD) (heart rate) model. 11-OH-THC, 11-hydroxy-THC; HR, heart rate; IV, intravenous.

**Figure 2 pharmaceutics-17-00237-f002:**
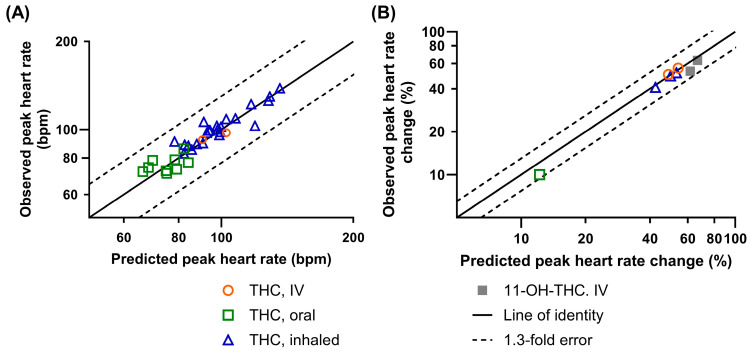
Observed vs. predicted peak heart rate (**A**) following THC administration in healthy adults by multiple routes of administration, and peak percentiles of heart rate change from the baseline (**B**) after THC administration to healthy adults by multiple routes of administration or following IV 11-OH-THC administration.

**Figure 3 pharmaceutics-17-00237-f003:**
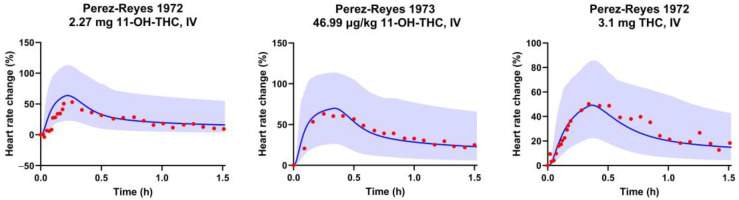
The observed and PBPK-PD model-predicted heart rate or percentage change in heart rate from the baseline. The blue shaded areas represent the 5th to 95th percentiles of the predicted values. The blue lines and red circles represent the mean predicted value and observed heart rates, respectively.

**Figure 4 pharmaceutics-17-00237-f004:**
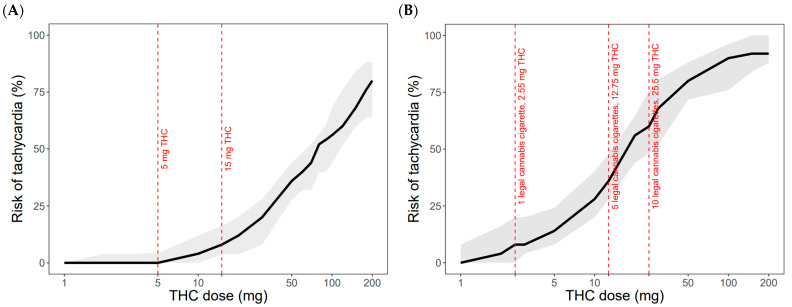
The PBPK-PD model-predicted risk of tachycardia at rest condition following oral (**A**) and inhaled (**B**) THC administration with a dose range from 1 mg to 200 mg. The grey shaded areas represent the 5th to 95th percentiles of the predicted risk of tachycardia at a rest condition. The black lines represent the mean predicted risk of tachycardia. The red dashed lines represent specific doses.

**Table 1 pharmaceutics-17-00237-t001:** The final input parameters for the delta-9-tetrahydrocannabinol (THC) absorption model, and the THC and 11-hydroxy-THC (11-OH-THC) pharmacodynamics model on heart rate.

Parameter	THC		11-OH-THC	
	Value	Reference	Value	Reference
**Absorption**				
Model Type	First-order		/	
f_a_	0.45	Qian 2025 [56]	/	/
k_a_ (h^−1^)	0.7	Qian 2025 [56]	/	/
Lung f_a_	0.22 ^a^, 0.05 ^b^, 0.4 ^b^, 0.5 ^b^, 0.6 ^b^, 0.9 ^b^	Qian 2025 [56]; Optimized	/	/
Lung k_a_ (h^−1^)	12 ^a^, 200 ^b^	Qian 2025 [56]; Optimized	/	/
**Pharmacodynamics**				
Model Type	Custom Lua model		Custom Lua model	
k_1e_ (/h)	4	Optimized	15	Optimized
k_e0_ (/h)	6.5	Optimized	15	Optimized
E_max_ (%)	105	Same as 11-OH-THC	105	Optimized
EC_50_ (μM)	0.15	Same as 11-OH-THC	0.15	Optimized
CV E_max_ (%)	32	Strougo 2008 [20]	32	Strougo 2008 [20]
CV EC_50_ (%)	79	Strougo 2008 [20]	79	Strougo 2008 [20]

f_a_, fraction absorbed from dosage form; k_a_, first-order absorption rate constant; lung f_a_, fraction of drug absorbed from the lung; k_1e_, rate constants for transport through the transfer compartment; k_e0_, rate constants for loss from the effect compartment; E_max_, the maximum fractional increase from the baseline heart rate; EC_50_, concentration in effect compartment required to obtain 50% of the maximum change; CV, coefficient of variation. ^a^ Parameter value from Qian 2025 [56]. ^b^ Parameter value optimized with THC concentration–time profiles. Details are provided in Appendix A and Appendix A.

## Data Availability

The data presented in this study are available in this article.

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
