# Peer review of "Quantifying Heart Rate Changes After Delta-9-Tetrahydrocannabinol Administration Using a PBPK-PD Model in Healthy Adults"

_pharmaceutics, 2025, doi:10.3390/pharmaceutics17020237_

Round 1

Reviewer 1 Report

Comments and Suggestions for Authors

The described by Qian  and Zhu Zhou model for assessing the tachycardia risk in both clinical and recreational cannabis use is important and valuable. The limitations of Authors’ model is also described very well. However, taking into account (1) the above numerous limitations of the new method, (2) the complex and still not fully understood mechanism responsible for THC-induced tachycardia and (3) the fact that most of the results regarding the various tachycardia mechanisms were obtained in rats in which basal HR is over 300 beats/min, while in humans only about 70 beats/min the Authors’ suggestions that their model enhances our understanding of the cardiovascular effects of THC is not fully convincing. Moreover, Authors have concentrated mainly on direct cardiac effect of THC which leads to tachycardia. However, the possibility that THC elicits tachycardia via direct activation of CB1-Rs in the sinoatrial node has not been demonstrated at all and it is not plausible, since these receptors are Gi/o protein-coupled, i.e., inhibitory and in the quoted by Authors publication by Maggo and Ashton [23; 2018] cannabinoid agonists (CP55940 and methanadamide had only moderate effects on isolated rat atria contraction rate. Moreover, the mentioned by the Authors cardiac mitochondrial cannabinoid CB1 receptors regulate mitochondrial respiration but not heart rate.

In this context I suggest that Authors should  concentrate mainly in their manuscript on detailed description and verification of  their PBPK-PD model as a valuable tool for prediction of changes in heart rate following intravenous, oral, and inhaled routes of THC administration without any suggestion that their model enhances our understanding of the cardiovascular effects of THC.

Author Response

  1. Reviewer comment: The described by Qian and Zhu Zhou model for assessing the tachycardia risk in both clinical and recreational cannabis use is important and valuable. The limitations of Authors’ model is also described very well. However, taking into account (1) the above numerous limitations of the new method, (2) the complex and still not fully understood mechanism responsible for THC-induced tachycardia and (3) the fact that most of the results regarding the various tachycardia mechanisms were obtained in rats in which basal HR is over 300 beats/min, while in humans only about 70 beats/min the Authors’ suggestions that their model enhances our understanding of the cardiovascular effects of THC is not fully convincing. Moreover, Authors have concentrated mainly on direct cardiac effect of THC which leads to tachycardia. However, the possibility that THC elicits tachycardia via direct activation of CB1-Rs in the sinoatrial node has not been demonstrated at all and it is not plausible, since these receptors are Gi/o protein-coupled, i.e., inhibitory and in the quoted by Authors publication by Maggo and Ashton [23; 2018] cannabinoid agonists (CP55940 and methanadamide had only moderate effects on isolated rat atria contraction rate. Moreover, the mentioned by the Authors cardiac mitochondrial cannabinoid CB1 receptors regulate mitochondrial respiration but not heart rate.

In this context I suggest that Authors should concentrate mainly in their manuscript on detailed description and verification of their PBPK-PD model as a valuable tool for prediction of changes in heart rate following intravenous, oral, and inhaled routes of THC administration without any suggestion that their model enhances our understanding of the cardiovascular effects of THC.

Response: 

We would like to thank the reviewer for the thoughtful and detailed comment. We appreciate your recognition of the importance of the model and its limitations. We agree that the mechanism behind THC-induced tachycardia is complex and not fully understood, particularly in humans, and we recognize the challenges in translating findings from animal models to humans. We have made substantial revisions following the feedback to address the reviewer’s concerns:

  1. Model Focus and Mechanisms of THC-Induced Tachycardia:
    We acknowledge that the mechanisms responsible for THC-induced tachycardia are complex and not fully understood, particularly in humans. As the reviewer pointed out, most data regarding these mechanisms come from animal studies (such as rats), which have much higher basal heart rates compared to humans. To clarify, we have revised the abstract and conclusion to remove any suggestion that our model enhances our understanding of THC's cardiovascular effects. Instead, we now emphasize that our model is a valuable tool for predicting heart rate changes after THC administration.
    • Abstract and Conclusion: We removed any statements suggesting that our model enhances understanding of THC's effect on heart rate, emphasizing its predictive value instead.
    • Introduction: We revised this section to focus solely on summarizing current hypotheses and studies regarding THC's effects on heart rate, without implying that our model offers new mechanistic insights. (Introduction: paragraph 3)
    • Discussion: We removed the detailed description of the mechanisms behind THC-induced tachycardia. Instead, we simply mention that the exact mechanism is not yet fully understood and remains an area for future research. (Discussion: paragraph 3)
  2. Direct Activation of CB1 Receptors in the Sinoatrial Node:
    We agree with the reviewer that the role of CB1 receptors in the sinoatrial node as a direct cause of tachycardia is unlikely, especially given their inhibitory (Gi/o-coupled) nature. In light of this, we have removed any overstatements regarding THC's direct cardiac effects. Instead, we focus on the model’s ability to predict heart rate changes based on the data we have, rather than speculating on specific mechanistic pathways. This better aligns with the reviewer’s comment and the current state of understanding.
  3. Emphasis on Model Verification and Validation:
    We have revised the manuscript to focus more on the verification and validation of our PBPK-PD model. We agree that the primary value of the model lies in its ability to predict heart rate changes after THC administration, rather than providing new insights into the mechanisms of THC’s cardiovascular effects. We have clarified the verification and validation process and re-emphasized the predictive capabilities of the model. This includes differentiating between the modeling, verification, and validation datasets, and highlighting how the model was tested and validated using multiple studies.

We hope these revisions clarify the scope and focus of our work and better reflect the current state of knowledge regarding THC’s cardiovascular effects. Once again, we would like to thank the reviewer for their insightful suggestions, which have significantly improved the clarity and scientific accuracy of our manuscript.

Reviewer 2 Report

Comments and Suggestions for Authors

Much needed work, with potential, but in my opinion requires significant modification or - maybe better word would be - clarification.

Major comments:

1 verification vs validation + concerns regarding workflow presented on Figure 1

In principle it is agreed that mathematical model verification consists of two elements:

code verification (determining whether the code correctly implements the intended algorithms)

and

solution verification (determining the accuracy with which the algorithms solve the mathematical model’s equations for specified endpoint of interest)

while validation is the process of determining the degree to which a computational model is an accurate representation of the real world from the perspective of the intended model applications (using other words – how good is it able to predict UNKNOWN DATA, without any fitting with the use of this data being simulated).

With that – have the models been VALIDATED? I assume no, and thus their extrapolation abilities are very limited, or at least unknown.

My question and statement can be further supported by the analysis of the supp table 1 – why various studies have different values of finh? Does that mean they were fitted for a each and every study separately? If so, this is really verification but the study lacks validation.

2 I strongly suggest sharing Simcyp workspaces and data used for model building as Supp materials for all readers. Otherwise, this work cannot be repeated, confirmed and properly reviewed.

3 The ke0, Emax, and EC50 were estimated à with the use of ALL data? For all studies at the same time? If so – there is no validation at all. If only subpart (1? 2? studies) were used to optimize and the rest was used to validate the optimized model, which studies were used?

BTW – this is how this should be done to follow good modelling practice!

4 The baseline heart rate was set at 70 bpm – why? how does this assumption influence study findings? Sensitivity analysis is very much needed.

5 ‘heart concentration’ à it does not really matter which concentration was used assuming perfusion limited model as in such a case Kp is just a linear scalar and fitting is used to correlate ANY concentration with effect

6 “which would require a clearly defined effect site” à it would much more require clearly defined affinity to transporters and thus distribution (and thus permeability limited PBPK Model)

7 “and we observed differences in these values, likely due to differences in lipophilicity and blood-to-plasma” à this is risky conclusion especially in the situation when there is no Sensitivity Analysis reported; fitting multiple parameters makes this model prone to be unidentifiable

8 “Finally, our model does not fully capture the entire mechanism of the THC effect on heart rate.” à This PD model does not capture mechanism at all. This is purely empirical model.

Minor comments:

1 Finh - I assume this is fraction of the drug which reached lung tissue, and NOT fraction of the drug which reached systemic blood flow after inhalation; if so - why capital 'F'; 

2 Interindividual variability of Emax and EC50 from a popPK model was applied to our heart rate model [20]. à this sentence is potentially misleading and should be rephrased.

3 𝐸max is the maximum increase fraction of 11-OH-THC on heart rate; à this sentence is potentially misleading and should be rephrased.

4 k1e is same as 𝑘e0 – why? Based on what observations?

5 inhaled THC from 1, 5, and 10 legal cannabis à from 1 through 5 till 10?

Author Response

  1. Reviewer comment: verification vs validation + concerns regarding workflow presented on Figure 1

In principle it is agreed that mathematical model verification consists of two elements:

code verification (determining whether the code correctly implements the intended algorithms) and solution verification (determining the accuracy with which the algorithms solve the mathematical model’s equations for specified endpoint of interest)

while validation is the process of determining the degree to which a computational model is an accurate representation of the real world from the perspective of the intended model applications (using other words – how good is it able to predict UNKNOWN DATA, without any fitting with the use of this data being simulated).

With that – have the models been VALIDATED? I assume no, and thus their extrapolation abilities are very limited, or at least unknown.

My question and statement can be further supported by the analysis of the supp table 1 – why various studies have different values of finh? Does that mean they were fitted for a each and every study separately? If so, this is really verification but the study lacks validation.

Response: Thank you for your valuable comment. We, too, have encountered confusion regarding the use of "verification" in many PBPK publications as they generally only use “verification” (for example, Shebley 2018, which differentiates qualification and verification, stating that "While qualification generally refers to a set of prerequisites that ensure “permission” to handle the intended use, verification, on the other, hand focuses on the predictive performance of the model"). Upon reviewing the FDA’s PBPK guidance, we found that it aligns with the definitions provided by the reviewer. While the use of these terms may cause some confusion within the PBPK community, we have now clarified both verification and validation in the Methods and Results sections of the manuscript per the reviewer’s suggestion. (Materials and Methods: 2.2 and 2.5)

We apologize for the earlier confusion regarding the modeling dataset. In the first version, all clinical studies used to develop the PD models and optimize inhalation absorption were labeled as the modeling group. In the latest version, we have separated these studies for clarity. Inhalation studies are now categorized into three groups in Figure S1 based on the availability of PK profiles and PBPK models, and we hope this categorization improves the transparency of our methodology.

The PD datasets are now correctly divided into modeling, verification, and validation datasets (Materials and Methods: 2.2 and 2.5):

  • The modeling dataset contains one clinical study using IV THC and 11-OH-THC, and parameter estimations were based on this study.
  • The verification dataset includes one inhaled and one oral THC clinical study with intensive heart rate sampling, used to verify the parameter estimates and model assumptions.
  • The validation dataset consists of IV and oral studies, as well as inhalation studies. No parameters were altered during validation for the IV and oral studies. For inhalation studies with PK profiles, only absorption parameters were optimized to capture the THC plasma concentration-time profiles, while PD parameters remained unchanged. For inhalation studies lacking concentration profiles, we used absorption parameters from other clinical studies to fit peak heart rate changes and assess predictive performance. No PD parameters were modified during validation, which ensures that the validation is focused on PD model performance.

Regarding the varying lung finh​ (now lung fa) values, these differences arise from the distinct inhalation processes and methods used in various studies. For example, in the Tashkin 1973 study, subjects inhaled THC from a cigarette over a 2-4 second period, holding their breath for about 15 seconds, while the Hunault 2009 study had a more complex procedure. Additionally, the Volcano® vaporizer was used in studies like Zuurman 2008, where THC was vaporized and stored in a balloon. These differences in inhalation methods naturally lead to variability in lung fa values.

Although inhalation absorption parameters were optimized to best predict peak heart rate changes, the developed exposure-response relationship allows for validation of predictive performance at other time points. We also acknowledge that this could be a limitation of our study: "Optimizations were required for inhaled THC administration to capture the PK profiles for different inhalation processes and methods. When the THC concentration-time profiles are absent in inhalation studies, optimizing inhalation absorption parameters based on peak heart rate changes may reduce the reliability of the validation." (Discussion: paragraph 4)

  1. Reviewer comment: I strongly suggest sharing Simcyp workspaces and data used for model building as Supp materials for all readers. Otherwise, this work cannot be repeated, confirmed and properly reviewed.

Response: Thank you for the suggestion. Since Simcyp workspaces require the Simcyp simulator to open and are sensitive to the software version, we have attached the Lua code for Simcyp simulator version 22 in the supplemental material. The mathematical equations and differential equations are already provided in the manuscript. The PBPK parameters for THC and 11-OH-THC are detailed in our previously published THC PBPK study. Additionally, the data we used were collected from various published sources.

  1. Reviewer comment: The ke0, Emax, and EC50 were estimated à with the use of ALL data? For all studies at the same time? If so – there is no validation at all. If only subpart (1? 2? studies) were used to optimize and the rest was used to validate the optimized model, which studies were used?

BTW – this is how this should be done to follow good modelling practice!

Response: Thank you for your comment. As mentioned in response to Comment 1, only one IV study with intensive heart rate sampling was used to estimate the ke0, Emax, and EC50 values. The estimates were then verified using one inhalation study and one oral study, both of which included intensive heart rate sampling. Additionally, the inhalation study also provided the THC plasma concentration-time profile, which ensured accurate prediction of THC PK.

  1. Reviewer comment: The baseline heart rate was set at 70 bpm – why? how does this assumption influence study findings? Sensitivity analysis is very much needed.

Response: Thank you for this valuable question. The baseline heart rate of 70 bpm was used only for the tachycardia risk simulation and was selected based on the latest guideline, “Guideline for the Application of Heart Rate and Heart Rate Variability in Occupational Medicine and Occupational Health Science.” According to the guideline, a resting heart rate for normal adults is recommended to be between 60 and 80 bpm. We chose the mean value of 70 bpm specifically for the simulation. To clarify this, we have added a sentence to the Methods section to indicate the source of the baseline heart rate for the risk simulation. (Materials and Methods: 2.6)

For all other model development, verification, and validation steps, the most recent heart rate measurement before THC dosing was used as the baseline whenever heart rate was reported in bpm. Since the model is sensitive to baseline heart rate, a sensitivity analysis that includes baseline heart rate was conducted for the tachycardia risk simulation (see Figure S3).

  1. Reviewer comment: heart concentration’ à it does not really matter which concentration was used assuming perfusion limited model as in such a case Kp is just a linear scalar and fitting is used to correlate ANY concentration with effect

Response: Thank you for your important question. We agree that, in the case of a single route of administration, the specific concentration used may not matter, as the Kp value acts as a linear scalar. However, during model development, we tested the effect of heart rate change driven by plasma, brain, and heart concentrations. Our findings indicated that only heart concentrations adequately described the observed heart rate changes following THC IV, oral, and inhaled administration. Models based on plasma concentrations underestimated the peak heart rate changes for inhalation, while those using brain concentrations underestimated heart rate changes for oral THC.

To further illustrate this, we examined the concentrations in the plasma, brain, and heart compartments after the three routes of administration (Figure S2). We found that peak plasma concentrations were closest to peak heart concentrations following IV and oral administration, whereas peak plasma concentrations were the lowest after inhalation. In contrast, brain concentrations were the highest and had the latest Tmax. The lower plasma concentration compared to heart concentration after inhalation is related to the organ of absorption: when THC is absorbed through the lungs, the pulmonary venous blood, rich in absorbed THC, first passes through the heart before being distributed to other parts of the body. For IV and oral administration, THC is distributed quickly in the blood and then flows through the heart. While the concentration used may not matter for a single route of administration, the differences observed across the various routes suggest that heart concentration provides the best results for modeling. We have added sentences to the Methods and Results sections to clarify our testing of the compartment driving the effect. (Materials and Methods: 2.4, paragraph 1)

  1. Reviewer comment: “which would require a clearly defined effect site” à it would much more require clearly defined affinity to transporters and thus distribution (and thus permeability limited PBPK Model)

Response: Thank you for your suggestion. THC is not known to be a substrate of transporters. Since our model is empirical, we believe it is beyond the scope of this work. As such, we have removed the related sentence from the manuscript.

  1. Reviewer comment: “Finally, our model does not fully capture the entire mechanism of the THC effect on heart rate.” à This PD model does not capture mechanism at all. This is purely empirical model.

Response: Thank you for your comment. We agree with the reviewer that our model is purely empirical and does not capture the underlying mechanisms. As a result, we have modified the sentence to: “Finally, our model does not include the mechanism of the THC effect on heart rate.” (Discussion: paragraph 4)

Minor Comments:

  1. Reviewer comment: Finh - I assume this is fraction of the drug which reached lung tissue, and NOT fraction of the drug which reached systemic blood flow after inhalation; if so - why capital 'F'

Response: Thank you for the suggestion, we have replaced Finh with lung fa, which is clearer and more commonly used. Additionally, we have revised the descriptions accordingly.

  1. Reviewer comment: Interindividual variability of Emax and EC50 from a popPK model was applied to our heart rate model [20]. à this sentence is potentially misleading and should be rephrased.

Response: We have revised the sentence to: "The interindividual variabilities of Emax and EC50 used in our model were derived from a published THC heart rate popPK/PD model". (Materials and Methods: 2.4, paragraph 2)

  1. Reviewer comment: ?max is the maximum increase fraction of 11-OH-THC on heart rate; à this sentence is potentially misleading and should be rephrased.

Response: Thank you for the comment. We have rephrased the sentence to “Emax is the maximum fractional increase from baseline heart rate induced by 11-OH-THC” (Materials and Methods: 2.4, paragraph 2)

  1. Reviewer comment: k1e is same as ?e0 – why? Based on what observations?

Response: Thank you for your insightful question. In a pharmacodynamic model with an effect compartment, it is common practice, as suggested by various textbooks and studies, to assume that k1e ​is equal to ke0​ to reduce the number of parameters to estimate. However, these sources also note that if this assumption does not hold, k1e​ and ke0​ can be treated as separate parameters. In our case, we optimized the model using studies with intensive heart rate sampling, which provided support for estimating different values for k1e ​and ke0​. For 11-OH-THC, the model with equal k1e ​and ke0​ accurately predicted the observations, so distinguishing between the two was unnecessary. However, for THC, different values for k1e and ke0​ were required to adequately capture the observations.

We have added the following sentence to the Methods section to clarify this approach: "The k1e and ke0 were assumed to be the same unless the optimized values could not capture the observations." (Materials and Methods: 2.4, paragraph 2)

  1. Reviewer comment: inhaled THC from 1, 5, and 10 legal cannabis à from 1 through 5 till 10?

Response: Thank you for your feedback. We simulated the risk of tachycardia following inhalation of THC, covering doses from 1 to 200 mg. To better contextualize real-world implications, we simulated and labeled three doses (1, 5, and 10 legal cannabis cigarettes) to represent varying levels of tachycardia risk based on different dosage amounts. We have rephrased the sentence for clarity. (Materials and Methods: 2.6)

Reviewer 3 Report

Comments and Suggestions for Authors

The manuscript provides a comprehensive study on the quantification of heart rate changes after the administration of Delta-9-Tetrahydrocannabinol (THC) in healthy adults using a physiologically based pharmacokinetic-pharmacodynamic (PBPK-PD) model. This research is highly relevant, particularly given the increasing legalization of cannabis and the concerns about THC-induced tachycardia. The integration of a PBPK-PD model, which incorporates both THC and its active metabolite 11-hydroxy-THC, offers valuable insights into the cardiovascular effects of THC and its potential risks of tachycardia in different dosing scenarios.

I recommend the manuscript for acceptance after addressing the following issues:

(1)    It would be beneficial for the authors to provide more detailed information on the model’s parameters, particularly the values used for THC absorption in inhaled administration. Expanding on the optimization process for these values would provide greater transparency and confidence in the model’s accuracy.

(2)    The authors mention that the placebo effect was not incorporated into the model due to a lack of data on control group heart rate changes. A discussion of how the placebo effect could influence the results, especially in a real-world scenario, would be valuable. Future models could account for such effects.

(3)    While the authors highlight the uniqueness of their approach, it would strengthen the paper to include a more detailed comparison with other existing pharmacokinetic models for THC. A comparative analysis would highlight the advantages and limitations of their model relative to previously published work.

(4)    In Equations (3) to (5), it would be helpful to provide a brief explanation of the terms used in the equations, particularly for readers who may not be familiar with pharmacokinetic modeling.

(5)    A sensitivity analysis regarding the influence of various parameters on the prediction of tachycardia risk would provide further insight into the robustness of the model and its potential limitations.

Comments on the Quality of English Language

 The manuscript is well-written, but minor revisions to improve sentence structure, clarity, and consistency would enhance its readability and impact.

Author Response

  1. Reviewer comment: It would be beneficial for the authors to provide more detailed information on the model’s parameters, particularly the values used for THC absorption in inhaled administration. Expanding on the optimization process for these values would provide greater transparency and confidence in the model’s accuracy.

Response: Thank you for your valuable suggestion. In response, we have revised the relevant paragraph in the Methods section to provide more detailed information on the model’s parameters, with a particular focus on the THC lung absorption parameter. (Materials and Methods: 2.3, paragraph 3) We hope that the revised section more clearly describes the optimization process and improves transparency, thereby enhancing confidence in the model’s accuracy.

  1. Reviewer comment: The authors mention that the placebo effect was not incorporated into the model due to a lack of data on control group heart rate changes. A discussion of how the placebo effect could influence the results, especially in a real-world scenario, would be valuable. Future models could account for such effects.

Response: Thank you for your valuable comment. We agree that discussing the potential impact of the placebo effect is important. In response, we have rephrased the relevant sentences in the Discussion section to explain our observations and the reasoning behind not including the placebo effect in the model. (Discussion: paragraph 4)

Given that the placebo effect on heart rate is highly variable, with changes that can either increase or decrease heart rate, we acknowledge that the influence of the placebo is not well-defined in our data. We cannot determine whether the heart rate changes observed in the control group are due to random fluctuations or are driven by some underlying cause. We also emphasize in the revised text that "The placebo effect in real-world scenarios should be considered" (Discussion: paragraph 5)

  1. Reviewer comment: While the authors highlight the uniqueness of their approach, it would strengthen the paper to include a more detailed comparison with other existing pharmacokinetic models for THC. A comparative analysis would highlight the advantages and limitations of their model relative to previously published work.

Response: Thank you for the suggestion. We added a more detailed comparison to the related paragraph in the discussion. (Discussion: paragraph 2)

  1. Reviewer comment: In Equations (3) to (5), it would be helpful to provide a brief explanation of the terms used in the equations, particularly for readers who may not be familiar with pharmacokinetic modeling.

Response: Thank you for bringing this to our attention. We have provided explanations for all the terms used in equations. (Materials and Methods: 2.4, paragraph 3)

  1. Reviewer comment: A sensitivity analysis regarding the influence of various parameters on the prediction of tachycardia risk would provide further insight into the robustness of the model and its potential limitations.

Response: We appreciate the reviewer's insightful suggestion. A sensitivity analysis is added. The results are shown in Figure S3.

  1. Reviewer comment: The manuscript is well-written, but minor revisions to improve sentence structure, clarity, and consistency would enhance its readability and impact.

Response: Thank you for your positive feedback and helpful comments. We have made revisions throughout the manuscript to improve sentence structure, clarity, and consistency, and we hope these changes enhance its readability and impact.

Round 2

Reviewer 1 Report

Comments and Suggestions for Authors

The authors have provided satisfactory replies to my comments and I have no further comments.

Author Response

Thank you!

Reviewer 2 Report

Comments and Suggestions for Authors

Almost there, just two minor comments:

1 𝑘1𝑒,11−𝑂𝐻−𝑇𝐻𝐶 is the rate constant from the compartment to the 11-OH-THC effect compartment --> rate of what? 

2 Currently available in vivo and in vitro data do not support the modeling of these mechanism processes. --> this sentence looks like cut in half; ...mechanisms of certain processes?

Author Response

  1. Reviewer comment: ?1?,11−??−??? is the rate constant from the compartment to the 11-OH-THC effect compartment --> rate of what? 

Response: We thank the reviewer for pointing this out. We have revised the sentence to “ is the rate constant for 11-OH-THC transport from the compartment driving heart rate changes to its effect compartment”; we also changed the similar sentence for THC to “ is the rate constant for THC transport from the compartment that drives heart rate changes to the THC effect compartment”.

  1. Reviewer comment: Currently available in vivo and in vitro data do not support the modeling of these mechanism processes. --> this sentence looks like cut in half; ...mechanisms of certain processes?

Response: Thank you for your comment. We have revised the sentence to: “Currently available in vivo and in vitro data do not support modeling either the mechanism underlying THC-induced heart rate changes or the development of tolerance to THC's pharmacodynamic effects.”

Reviewer 3 Report

Comments and Suggestions for Authors

The author has almost addressed my questions and concerns in the revised submission. I have no further comments.

Author Response

Thank you!